# Re-Emergence of Rift Valley Fever Virus Lineage H in Senegal in 2022: In Vitro Characterization and Impact on Its Global Emergence in West Africa

**DOI:** 10.3390/v16071018

**Published:** 2024-06-25

**Authors:** Ousseynou Sene, Samba Niang Sagne, Ndeye Sakha Bob, Moundhir Mhamadi, Idrissa Dieng, Aboubacry Gaye, Haoua Ba, Moussa Dia, Elisabeth Thérèse Faye, Sokhna Mayemouna Diop, Yoro Sall, Boly Diop, Mamadou Ndiaye, Cheikh Loucoubar, Etienne Simon-Lorière, Anavaj Sakuntabhai, Ousmane Faye, Amadou Alpha Sall, Diawo Diallo, Ndongo Dia, Oumar Faye, Moussa Moise Diagne, Malick Fall, Marie Henriette Dior Ndione, Mamadou Aliou Barry, Gamou Fall

**Affiliations:** 1WHO Collaborating Centre for Arbovirus and Viral Hemorrhagic Fevers, Virology Department, Pasteur Institute, Dakar 12900, Senegal; ndeye.bob@pasteur.sn (N.S.B.); idrissa.dieng@pasteur.sn (I.D.); haouaba2@gmail.com (H.B.); moussa.dia@pasteur.sn (M.D.); elisabeththeresefaye@gmail.com (E.T.F.); sokhna.maymouna.diop.97@gmail.com (S.M.D.); ousmane.faye@pasteur.sn (O.F.); amadou.sall@pasteur.sn (A.A.S.); ndongo.dia@pasteur.sn (N.D.); oumar.faye@pasteur.sn (O.F.); moussamoise.diagne@pasteur.sn (M.M.D.); marie.ndione@pasteur.sn (M.H.D.N.); gamou.fall@pasteur.sn (G.F.); 2Epidemiology, Clinical Research & Data Science, Pasteur Institute, Dakar 12900, Senegal; sambaniang.sagne@pasteur.sn (S.N.S.); aboubacry.gaye@pasteur.sn (A.G.); cheikh.loucoubar@pasteur.sn (C.L.); aliou.barry@pasteur.sn (M.A.B.); 3DIATROPIX, Pasteur Institute, Dakar 12900, Senegal; moundhir.mhamadi@pasteur.sn; 4Ministry of Health, Dakar 10700, Senegal; yorosall2005@yahoo.fr (Y.S.); diopboly@yahoo.fr (B.D.); mamamorph@yahoo.fr (M.N.); 5G5 Evolutionary Genomics of RNA Viruses, Pasteur Institute, 75015 Paris, France; etienne.simon-loriere@pasteur.fr; 6Functional Genetics of Infectious Disease Unit, Pasteur Institute, 75015 Paris, France; anavaj.sakuntabhai@pasteur.fr; 7Centre National de la Recherche Scientifique (CNRS), UMR2000, Department of Global Health, 75015 Paris, France; 8International Vaccine Design Center (vDesC), the Institute of Medical Science, University of Tokyo (IMSUT), Tokyo 108-8639, Japan; 9Zoology Department, Pasteur Institute, Dakar 12900, Senegal; diawo.diallo@pasteur.sn; 10Animal Biology Department, Faculty of Sciences and Techniques, Cheikh Anta Diop University, Dakar 10700, Senegal; malick.fall@ucad.edu.sn

**Keywords:** Rift Valley fever, re-emergence, lineage H, in vitro characterization, Senegal, 2022

## Abstract

Rift Valley fever (RVF) is a re-emerging vector-borne zoonosis with a high public health and veterinary impact. In West Africa, many lineages were previously detected, but since 2020, lineage H from South Africa has been the main cause of the outbreaks. In this study, clinical samples collected through national surveillance were screened for RVF virus (RVFV) acute infection by RT-PCR and IgM ELISA tests. Sequencing, genome mapping and in vitro phenotypic characterization in mammal cells were performed on RT-PCR positive samples in comparison with other epidemic lineages (G and C). Four RVFV human cases were detected in Senegal and the sequence analyses revealed that the strains belonged to lineage H. The in vitro kinetics and genome mapping showed different replication efficiency profiles for the tested RVFV lineages and non-conservative mutations, which were more common to lineage G or specific to lineage H. Our findings showed the re-emergence of lineage H in Senegal in 2022, its high viral replication efficiency in vitro and support the findings that genetic diversity affects viral replication. This study gives new insights into the biological properties of lineage H and calls for deeper studies to better assess its potential to cause a future threat in Senegal.

## 1. Introduction

Rift Valley fever (RVF) is an emerging and re-emerging vector-borne zoonosis, first discovered in the Rift Valley Province of Kenya in 1931 [1,2]. The disease is caused by a mosquito-borne virus named Rift Valley fever virus (RVFV) belonging to the *Phlebovirus* genus of the *Phenuiviridae* family [3]. RVFV can be transmitted to animals and humans through bites of infected mosquitoes mainly of the genera *Aedes* and *Culex* [4]. The virus can be transmitted by horizontal transmission in vertebrates and by transovarial and/or vertical transmission in mosquito eggs during periods of drought. The main modes of transmission to humans include exposure to body fluids, blood and tissues of infected animals or contact with aerosols. Human-to-human spread has not been documented [5]. In animals, the disease is characterized by abortions among pregnant females and high mortality rates for offspring [1,6]. For human infections, symptoms are generally mild (Flu-like syndrome with fever (38–40 °C), headache, muscle pain, weakness, nausea and epigastric discomfort and photophobia) but may evolve into severe symptoms, such as hemorrhage, meningoencephalitis and retinopathy with fatal outcomes [5,7]. Recently, RVFV infection has also been associated with spontaneous abortion among pregnant women in Sudan [8].

RVFV is characterized by three single-stranded RNA genomes: L (large), M (medium) and S (small). The negative sense L segment encodes the RNA-dependent RNA polymerase (L) gene. The negative sense M segment encodes a non-structural protein (NSm) and glycoprotein (Gn and Gc) genes. The S segment encodes the nucleoprotein (N) and non-structural (NSs) genes in an ambisense manner; NSs play a major role in innate immunity and interact with interferon signaling pathways [9,10]. A previous study of 198 isolates obtained over a 67-year period from numerous countries classified sequences into 15 lineages, A to O [11]. Among them, lineages A, C, G, K and N were previously reported in West Africa, but in 2020, all the reported cases were due to lineage H from South Africa and Namibia [11,12,13]. Genetic characterization suggests that all the strains are closely related, but there are some regional differences, suggesting two or three regional virus types [14].

RVF outbreaks have so far been confined to Africa and the Arabian Peninsula, but many potential RVFV vectors can be found outside these two geographical settings [4]. Recently, a new RVF case was reported in China (Asia), imported from Angola [15]. Therefore, RVFV is of major concern to public health. Indeed, since 2015, RVF has been part of the Research & Development (R&D) Blueprint program of the World Health Organization (WHO), a list of top emerging diseases likely to cause major epidemics and for which few or no medical countermeasures exist [16]. Moreover, RVF causes huge losses in livestock that can dramatically impact food production locally and hamper exports. All this could cause a severe socioeconomic impact on both animals and humans [14]. Strategies for RVFV control and eradication should, therefore, be implemented using One Health concepts [4].

In West Africa, particularly in Mauritania and Senegal, there was an epidemic associated with the creation of a dam on the Senegal River in 1987–1988 that caused 220 deaths in humans [7,17]. Since then, there has been a constant circulation of RVFV in Mauritania with major outbreaks and deaths in 1987, 2010, 2012, 2015, 2020 and, most recently, in 2022 when 47 RVF human cases, including 23 deaths, were recorded [18]. However, in Senegal, despite regular detections of RVF in livestock and mosquitoes, only sporadic cases or minor outbreaks have been notified [13,19,20]. Recently, a survey conducted in the Matam region (Northern Senegal) showed high seroconversion with RVFV IgG antibodies in sheep (66.67%) and humans (20.78%) [21], suggesting high circulation of the virus in this area located at the border with Mauritania. Indeed, during the last RVF outbreak in Senegal in 2020, six out of the nine confirmed cases were reported in Northern Senegal (Saint-Louis and Matam regions), indicating the high circulation of RVFV in this area. Many factors such as rainfall, high temperatures, vector ecology, host genetics, viral genetic diversity and livestock movements during religious events could explain the emergence and re-emergence of RVFV in both countries as well as the different epidemiological patterns.

In this study, we describe four new cases of RVF detected in Saint-Louis (Northern Senegal), Matam (Northeast Senegal) and Dakar (Western Senegal) by the sentinel syndromic surveillance system network (4S) during the year 2022 in Senegal. Molecular and phenotypic characterizations of the strains were also investigated.

## 2. Materials and Methods

### 2.1. Presentation of the Syndromic Sentinel Surveillance System Network in Senegal (4S Network) and Sample Collection

Since 2011, the Prevention Department (Epidemiological Surveillance Division) of the Ministry of Health and Social Action (MSAS), in collaboration with the Pasteur Institute of Dakar (IPD), has set up a surveillance system known as the “Syndromic Sentinel Surveillance System” or “4S Network”. This 4S network is a complementary tool to a routine surveillance system that is already in place to identify epidemic risk earlier and respond more quickly [13,22]. This 4S system relies on standard case definitions recommended by the WHO and focuses on febrile illnesses of particular public health importance in Senegal. Regarding arbovirus surveillance, patients are considered suspect if they present with a fever of ≥38 °C along with at least two of the following symptoms: headache, joint pain, muscle pain, rash, retro-orbital pain, encephalitis or hemorrhagic symptoms. In the case of these suspect patients, comprehensive individual records are meticulously filled out, encompassing sociodemographic, clinical and paraclinical details (including results from malaria rapid diagnostic tests). Additionally, blood samples are collected and sent to the IPD for virological analysis.

### 2.2. Analysis of Samples Collected through the 4S Network

#### 2.2.1. Serological Testing

At the IPD, the samples were tested using in-house IgM ELISA tests for different viruses, including RVFV, using antigens and immune ascites produced in mice at IPD, as previously described [13].

#### 2.2.2. RT-PCR Testing

RT-PCR tests were also conducted on the collected blood samples for different viruses, including RVF, as already described [13]. Briefly, viral RNA was extracted from a 140 µL blood sample using the QIAamp viral RNA Mini kit (QIAGEN, Hilden, Germany) according to the manufacturer’s instructions (Qiagen, Hilden, Germany). A QuantiTect probe (Qiagen, Hilden, Germany), primers and a previously described probe were used for RT-PCR [23].

#### 2.2.3. Viral Genome Sequencing and Bioinformatics Analyses

Sequencing was performed using the RNA extracted from the 2 RVF RT-PCR positive samples. Initially, the host ribosomal RNA was depleted using specific probes and Oligo-dT. The mixture underwent heat denaturation at 95 °C for 2 min, followed by slow cooling to 45 °C at a rate of 0.1 °C/s. Subsequently, RNA hybrids were degraded using RNase H (NEB), followed by treatment with TURBO DNase (ThermoFisher, Waltham, MA, USA) and purification using RNAClean XP beads (Beckman Coulter, Brea, CA, USA). The depleted RNA served as a template for first-stranded cDNA synthesis using the SuperScript IV Reverse Transcriptase kit (Invitrogen, Thermo Fisher, Waltham, MA, USA). The double-stranded cDNA was generated using the Klenow exo-DNA polymerase (NEB, Hitchin, UK). Whole-genome sequencing was performed following a hybrid capture approach with the Twist Biosciences Comprehensive Viral Research Panel (CVRP). Briefly, cDNA was fragmented before a telomere repair, a dA-Tailing and a ligation step with Universal Twist adapters before a final library amplification, as previously described [24]. A single library pool was finally prepared from the indexed library-prepped samples and then subjected to target hybridization in solution, followed by binding of the hybridized targets to the desired streptavidin beads. The enriched sample libraries, obtained as recommended by Twist Bioscience Technical Support, were loaded onto an Illumina iSeq100 instrument (San Diego, CA, USA).

The sequencing reads were analyzed using the EDGE bioinformatics pipeline (https://www.edgebioinformatics.org/, accessed on 12 December 2022). Nearly complete genomes were submitted to the public database BLAST to identify homologous sequences. Subsequently, they were combined with a representative subset of RVFV sequences available in GenBank. To align the sequences, the Mafft program [25] was used. Maximum Likelihood (ML) phylogenetic trees were performed using IQ-Tree software [26] by adding an automatic model selection argument using a model finder (MF) implemented on the software using the Bayesian Information criterion and were tested using the bootstrap method with 1000 replicates. The trees were visualized and annotated using Figtree [27].

### 2.3. RVFV In Vitro Replication Kinetics

#### 2.3.1. Virus Stains

Phenotypic characterization was conducted in vitro for a new RVFV strain belonging to lineage H in comparison with known reference strains circulating in West Africa and available in the Biobank of the WHO collaborating Center for Arbovirus and Hemorrhagic Fever Viruses (CRORA) in the Virology Department of IPD (Table 1). Lineage C, which was responsible for the outbreak in Mauritania in 2003, with 25 cases including 16 hemorrhagic signs and 4 deaths, was included in the study [28]. The ZINGA strain, responsible for the endemicity of RVFV in Central Africa from 1976 to 1986, with nine cases and three deaths associated with hemorrhagic signs [29], was also used (Table 1).

#### 2.3.2. Cells Lines

Vero cells (African green monkey kidney epithelial cells; *Cercopithecus aethiops*) and PS cells (porcine stable kidney cells, ATCC number, Manassas, VA, USA) obtained from Sigma Aldrich, France, were grown in L15 (Leibovitz’s 15) medium supplemented with 10% heat-inactivated fetal bovine serum (FBS), 1% penicillin-streptomycin, 0.05% amphotericin B (Fungizone) (GIBCO by Life Technologies, CA, USA) and 10% tryptose phosphate (Becton, Dickinson and Company Sparks, NJ, USA). The cells were incubated at 37 °C without CO_2_.

#### 2.3.3. Viral Stock Preparation

Vero cells were cultured in a T25 cm^2^ cell culture flask (BD Falcon, Erembodegem, Belgium) and incubated at 37 °C without CO_2_ until they reached approximately 80% confluence. The medium was removed and 200 µL of viral solution was added to the cells. After 1 h of incubation, 5 mL of L15 medium (5% FBS, 1% penicillin-streptomycin and 0.05% amphotericin B) was added, and the infected cells were then incubated for 4 days until extensive cytopathic effects could be observed. The supernatants were tested using RT-PCR and quantified using a plaque-forming unit (PFU) assay on the PS cell culture, as previously described [23,30,31,32]. The supernatants of the infected cells were aliquoted, frozen at −80 °C and used as viral stocks for growth kinetics.

#### 2.3.4. Growth Kinetics

For each strain, experiments of growth kinetics were conducted in triplicate, as previously described [30,32]. As data were not available for RVFV growth kinetics, we followed the methods that were previously used for West Nile and Yellow Fever viruses in our laboratory [30,32]. Briefly, Vero cells were infected in 12-well plates using one viral strain per plate, and an uninfected well serving as a negative control. Each well was initially seeded with 2.4 × 10^5^ cells in a 400 µL volume of the appropriate cell culture medium and then infected with 2.4 × 10^3^ PFU of virus in 400 µL of medium, resulting in a multiplicity of infection (MOI) of 10^−2^. After a 4 h incubation period, the medium was replaced with 2 mL of fresh cell culture medium to establish the starting point for the growth curves (T0). Supernatant harvesting was performed at 22, 28, 50, 75, 99, 124 and 146 h post-infection (pi). RNA was extracted from the supernatants collected for each strain at different times pi and then analyzed by RT-PCR, as previously described [23], to estimate the number of total viral particles released along the kinetics experiments. To quantify the infectious viral particles among the total viral particles released at different times for each strain, the supernatants were also titrated using plaque assay tests, as previously described [30]. Finally, the replication efficiency of the strains was assessed by calculating the ratio of the total number of particles released (estimated by RT-PCR) to the number of infectious particles (estimated by titration) for each time point [33]. This replication efficiency could be indicative of the virulence of the different strains in vivo.

The growth rates were compared using R software (R version 4.3.3, The R Foundation for Statistical Computing). The Wilcoxon (Mann–Whitney) test, which permits a comparison of strain replication in pairs at each sampling time (significant when the *p*-value was less than 0.05), was also conducted.

#### 2.3.5. Quantification of Standard RVFV Viral RNA

The RVFV RNA of the supernatants collected at different time points of the growth kinetics were quantified using an RVFV in vitro RNA (ivRNA) targeting the N gene in the segment S. This iv RNA was generated at GenExpress (Berlin, Germany) by inserting the ligated amplicons of 517 bp covering the targeting RT-qPCR region of RVFV into the plasmid pCRII (Life Technologies, GmbH, Darmstadt, Germany). The plasmid was then generated and transcribed to RNA by TibMolBiol (Berlin, Germany) at a concentration of 10^10^ RNA molecules/reaction according to the manufacturer’s recommendations. This ivRNA (standard RNA) was purchased from the TibmolBiol company and shipped to our laboratory in a dry format.

Ten-fold dilutions of the standard RNA, 10^6^ to 0.1 genome copy numbers, were quantified in triplicate using the RVFV primers and probes as described in [23]. Regression curves were obtained representing the RNA copy number/reaction vs. the threshold cycle value (Ct). The lowest RNA copy number detectable by the RT-PCR test was considered the analytical detection limit. The amplification efficiency of the primers was calculated from the slope of the standard regression lines, E = 10^(−1/slope)^ − 1. The genome copy numbers in the supernatants were calculated using the quantification curve of the ivRNA.

#### 2.3.6. Map and Genomic Analysis

Similarity plot (SIMPLOT) v. 3.5.1 was used to further analyze the genetic variations in the S, M and L segments between lineage H and the other strains used for the in vitro kinetics [34].

As the NSs protein is considered the major virulence factor for RVFV, we also performed an alignment with different strains to detect mutations that could be correlated to the phenotypes in vitro [35,36]. Indeed, we used the NSs protein of the strains SH172805 corresponding to lineage C and SH409226 corresponding to lineage H. For lineage G, as NSs is not available for the ZINGA strain, we used the Guinea 1981 strain also belonging to lineage G [11]. Finally, we also included other lineage H strains (two epidemic strains isolated in South Africa in 2010, Mauritania in 2020 and Senegal in 2020) to analyze the polymorphism within lineage H. These alignments were performed using Mega 7 software [37].

## 3. Results

### 3.1. Analysis of Samples Collected through the 4S Network

During 2022, 2368 samples were sent to IPD for testing, among them 4 RVF-confirmed cases were detected, 2 RT-PCR and 2 IgM-positive samples (Table 2). All the patients recovered without complications or sequelae. RVF-confirmed patients had a mean age (19 to 56) of 35 years and a M/F sex ratio of 1. Among these cases, two were located in the Matam region, one case in the Saint-Louis region and one case in the Dakar region (Figure 1).

### 3.2. Phylogeny

BLASTn analyses revealed that the 2022 strains shared very high nucleotide sequence similarities (more than 98%) with strains previously identified during 2009 and 2010 RVFV outbreaks in South Africa and isolates detected in Namibia in 2004 and 2010. Phylogenetic analysis based on the L, M and S revealed that our strains clustered within genetic lineage H with West African strains, specifically found in Senegal and Mauritania in 2020, as well as the epidemic strains found in Namibia in 2010 and South Africa in 2009–2011 (Figure 2) [38,39,40,41]. Moreover, the phylogenetic trees with all the segments showed that our new sequences are more closed to the previous lineage H strains identified in Senegal in 2020 (SH326064, SH327052, SH328056) [13].

### 3.3. RVFV Replication Kinetics

#### 3.3.1. Validation of the Standard RVFV Viral RNA

The analytical detection limit of the RT-PCR assay tested using the RNA standard was 100 copies/reaction (Figure A1, Appendix A). The standard RNA equation y = 3.7008x + 15.85 allowed for quantification of the number of genome copies for the RVFV strains characterized in this study.

#### 3.3.2. Viral Stocks

For the in vitro kinetics experiments, since the two new strains belonging to lineage H were very closed, only one was chosen for the phenotypic characterization in comparison with other reference RVFV strains (lineages C and G). The viral titers for the different strains were 2.5 × 10^7^ pfu/mL, 5.7 × 10^7^ pfu/mL and 7.5 × 10^7^ pfu/mL, respectively, for SH172805 (lineage C), SH409226 (lineage H) and ZINGA (lineage G).

#### 3.3.3. Growth Kinetics

The analysis of the number of total viral particles released during the kinetics showed differences in genome replication dynamics among the different RVFV strains. Indeed, viral amplification and the release of infectious particles started for lineage C at 22 h post-infection (hpi), while lineage H initiated viral particle release at 50 hpi, followed by lineage G at 75 hpi (Figure 3). These data showed an early start of the viral replication for lineage C, which was delayed for lineages H and G.

Regarding the analysis of the genome copy numbers in the supernatants, corresponding to the total number of released particles, the three lineages had different profiles at the early stages of the growth kinetics, with lineage C showing the highest genome copy numbers up to 50 hpi (*p*-values ranging from 0.003 to 0.010) and comparable profiles at the late stages of the growth kinetics, from 75 to 148 hpi (Figure 3, Panel A).

Regarding the analysis of the viral titers in the supernatants, corresponding to the number of infectious particles, lineage C showed the highest numbers from 22 to 75 hpi and lower numbers from 124 to 146 hpi (*p*-values ranging from 0.009 to 0.014) than lineages G and H, which were comparable. (Figure 3, Panel B).

Finally, for the viral replication efficiency, all the RVFV strains analyzed had a good replication efficiency, producing as many copy genomes as viral infectious particles. Lineage C showed the highest replication efficiency at the early stage of the kinetics, up to 28 hpi (*p*-values ranging = 0.0369). Between 50 and 75 hpi, all the strains showed comparable replication efficiencies, while from 99 to 148 hpi, lineages H and G showed the highest replication efficiencies (*p*-values ranging from 0.005 to 0.01) (Figure 3, Panel C).

#### 3.3.4. Map and Genomic Analysis

SIMPLOT analyses revealed high similarity (about 94–100%) between the different lineages along the genome, and more variations were observed in Gn, Gc and NSs proteins. The results also showed that lineage H was more closed to lineage G (Figure 4, Panel A).

The alignment showed, using lineage C as a reference, five non-synonymous mutations, leading to amino acid changes in the NSs protein (Figure 4, Panel B). Regarding the three lineages used for the growth kinetics, we identified two conservative amino acid mutations, with one shared between lineages G and H at position 141 (valine to isoleucine) and a second one specific to lineage H at position 214 (alanine to glycine). Interestingly, we also found two non-conservative amino acid mutations at position 245 (isoleucine to threonine) that are specific to lineage H and at position 253 (glycine to glutamic acid) that are common between lineages G and H. Finally, between the lineage H strains, mutations were also detected. Indeed, compared to the strain from South Africa in 2010, the two non-conservative mutations at positions 245 and 253 seem to be specific to the lineage H strains isolated in Senegal in 2020 and 2022.

## 4. Discussion and Conclusions

Our study reports four new RVFV cases detected in the Matam, Saint-Louis and Dakar regions in 2022 using the 4S network. These regions are characterized by significant hydrographic networks, including rivers and wetlands, which contribute to their abundance of natural resources. Interestingly, the close geographic proximity of Matam and Saint-Louis in Northern Senegal to Mauritania, where RVFV is highly endemic, strengthens cross-border connections, facilitating trade and population mobility. Furthermore, there is an important livestock exchange between Mauritania and Senegal, especially during traditional major Muslim holidays. Finally, Mauritania as well as Northern Senegal offer environmental conditions favorable to the RVFV persistence in nature and are marked by the presence and abundance of RVFV-competent vectors [42,43]. Then, the detection of the new cases in 2022 corroborates previous data and confirms RVFV endemicity in Northern Senegal. However, as RVFV endemicity has not yet been proven in Dakar, the RVFV case detected in Mariste could probably be due to travel or contact with animals from endemic areas.

The detection period for these cases spans the months of August to November, corresponding with the mid- and late-rainy seasons in Senegal. During this period, rainfall frequency affects the dynamics of competent RVFV vector populations, particularly their flight range [43]. Interestingly, in 2022, Senegal experienced significant climatic disturbances, marked by exceptionally heavy rains, and the resultant flooding, as previously shown, likely facilitated the massive emergence of RVFV-competent vectors [44,45]. Moreover, RVF outbreaks have expanded geographically in Senegal, with mainly the same areas being affected during the recent outbreaks [13,46]. Indeed, six out of the nine RVFV cases detected in Senegal in 2020 were also from Northern Senegal [13]. This correlation between the detection period, the climatic disturbances and the successive outbreaks in the same localities suggests an important role of the mosquito vectors in the transmission of RVFV in these areas. This data also indicates high circulation of the virus in the affected regions, mostly located in the northern part of Senegal. Indeed, for a long time, Northern Senegal has been recognized as a hotspot for RVFV emergence, with studies revealing that Barkedji in the Louga region functions as a hub, broadcasting RVFV to other locations in West Africa [12]. This calls for the enhancement of the surveillance in Northern Senegal using the One Health approach.

All of these patients presented common signs of an infectious syndrome, such as headaches (100%), muscle pains (75%) and joint pains (50%) and were detected at the early stage of the infection, which facilitated their clinical management until remission without complications. This underlines the importance of a surveillance system like the 4S network for early case detection, mainly by RT-PCR and the rapid management of RVFV cases. This also offers the possibility of genomic surveillance of RVFV to better characterize the circulating strains, which could have different biological properties as well as severity in humans. Based on the maximum likelihood trees, our new strains clustered within the genetic lineage H with West African strains, specifically found in West Africa in 2020 (Senegal and Mauritania) as well as epidemic strains isolated in Namibia and South Africa between 2009 and 2011 [38,39,40]. Moreover, the high similarity between these new RVFV sequences and the previous ones detected in 2020 in Senegal, specifically in Ndiaye-Ndiaye (Fatick Region) and Bokidiawe (Matam Region) as well as in Assaba (Mauritania) at the border with Senegal, suggests the re-emergence of lineage H in Senegal in 2022 [13,41]. Overall, these findings suggest the introduction of the RVFV lineage H into West Africa from Southern Africa and its subsequent diffusion in West African countries, as already shown for previous RVFV lineages in West Africa [12]. However, as there is a lack of genomic data on the RVF strains responsible for the recent outbreak in Mauritania in 2022 as well as in other countries in West Africa, further studies are needed to better estimate lineage H spreading in West Africa. This introduction of RVFV lineages into the West African region from Southern Africa as well as their diffusion in West Africa probably occurs through the livestock trade or the movements of infected mosquitoes or humans.

Previous data supported that the same RVFV lineage H strain initially found in Namibia in 2004 caused later large-scale epidemics (241 human cases including 25 deaths) as well as epizooties (more than 14,000 animal cases including 8000 deaths) in South Africa in 2010 [40,47,48]. Subsequently, lineage H also caused an RVFV outbreak in Namibia in 2010 where analysis of animal specimens confirmed virus circulation on seven farms [39]. In Mauritania, lineage H was also responsible for an outbreak involving 78 human cases and 186 animal cases [41]. The severity of the human cases, their large spatial distribution, the high number of deaths (25 humans) and the specific severe signs observed for the first time in camels (blindness and hemorrhagic syndrome) were remarkable during this lineage H outbreak [41]. All these data in Namibia, South Africa and Mauritania suggest high virulence for lineage H. However, only a few minor sporadic cases were reported in Senegal in 2020 and 2022 and no death in humans has been reported [13]. This epidemiological profile in Senegal, showing a minor impact on RVF, was also observed with the previous lineages detected between 2013 and 2020 [13,46]. This profile in Senegal could be due to many factors related to humans, vector competence and the environment that are not yet favorable enough to lead to a major RVFV outbreak in Senegal. This could also be due to the lower virulence of this particular lineage H strain detected in 2020 and 2022 in Senegal. Indeed, it has been shown for other arboviruses such as West Nile virus that mutations between strains could lead to the existence of viral strains with high and low virulence that could exist within the same lineage [32]. Interestingly, we found non-conservative mutations specific to the lineage H strains isolated in Senegal in this study and this could lead to different viral replication efficiencies and virulence within lineage H.

Therefore, we explored the viral replication efficiency of the RVF lineage H strain detected in Senegal. However, as we do not have the epidemic and virulent lineage H strains from Mauritania, Namibia or South Africa, we compared our strain with known epidemic strains belonging to lineage C and G. Phenotypic characterization in vitro in mammal cells revealed that lineage C had a shorter viral life cycle than the lineage H and G strains and more efficient viral replication at the early stages of the kinetics. However, during the mid and late stages of the kinetics, lineage H was comparable with lineage G and showed the highest replication efficiency. These data showed different replication efficiency profiles for the different RVFV strains, in a time-dependent manner, and support the findings that viral genetic diversity most likely affects viral cycle replication and virulence, as previously described [32,49,50]. Indeed, we found genome variations along the genome and more diversity in the glycoproteins involved in the viral entry into host cells [50]. In addition, focusing on the NSs protein, non-conservative mutations (hydrophobic to hydrophilic amino acids) were found between the different strains and could most likely result in a change in the protein’s shape or function. As it has been shown that the NSs are the major virulence factor of Bunyaviruses, including RVFV, and play a crucial role in viral replication, these mutations could lead to the different viral replication profiles we observed in the growth kinetics study [35,36]. Concordantly, lineages H and G, which shared more mutations in the NSs, showed more comparable viral replication efficiency along the growth kinetics study compared to lineage C. However, other proteins such as the L, NSm and glycoproteins also play crucial roles in RVFV viral replication; therefore, more studies are needed to decipher all the non-conservative mutations that regulate RVFV viral replication and virulence.

Our study suggests high and comparable replication efficiency in vitro in mammal cells for all these strains, including the lineage H strain from Senegal. As previously described, lineages C and G were responsible for severe outbreaks in Africa, particularly in Mauritania and Central Africa, with numerous deaths [28,29]. Therefore, a large spreading of this lineage H strain in Senegal, particularly in areas where RVFV vectors and rainfall are very abundant, could lead to major outbreaks in the future.

Further studies in vivo using mice models and other RVFV lineages, including the virulent lineage H strains from Mauritania or other countries (Namibia or South Africa), are needed to better explore and understand the virulence of the RVFV lineage H strain detected in Senegal and its link with the viral genetic diversity. Vector competence studies are also needed to assess the capacity of the RVFV vectors present in Senegal to transmit this new lineage. These studies could allow a better understanding of the potential of this lineage H strain to cause future threats in Senegal.

This study showed the importance of syndromic sentinel surveillance in the detection of RVFV cases and provided new insights into the biological properties of the lineage H strain detected in Senegal. It also showed the importance of pathogen genomic surveillance to detect genotypes and predict phenotypes, which will ensure the timely identification of infections with high epidemic potential and help refine control strategies.

## Figures and Tables

**Figure 1 viruses-16-01018-f001:**
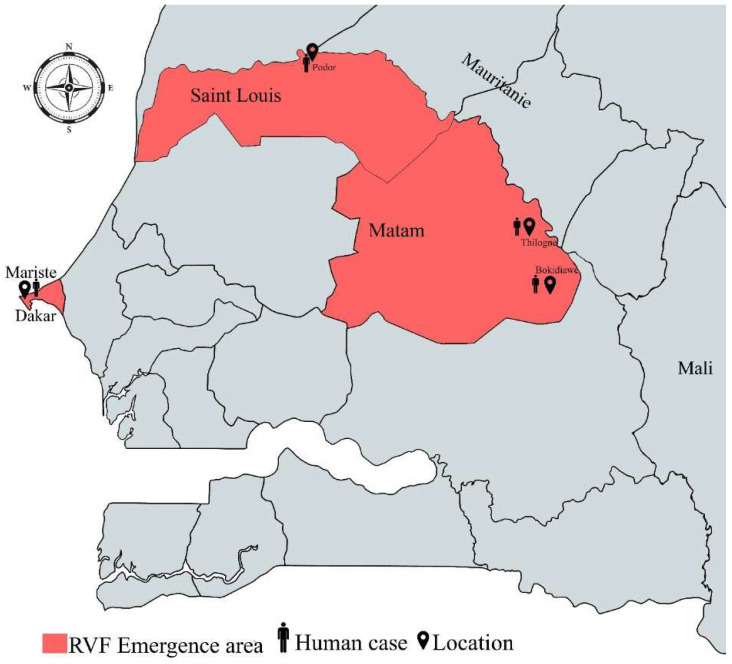
Geographical representation of RVF emergence areas (
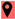
 location of RVF cases).

**Figure 2 viruses-16-01018-f002:**
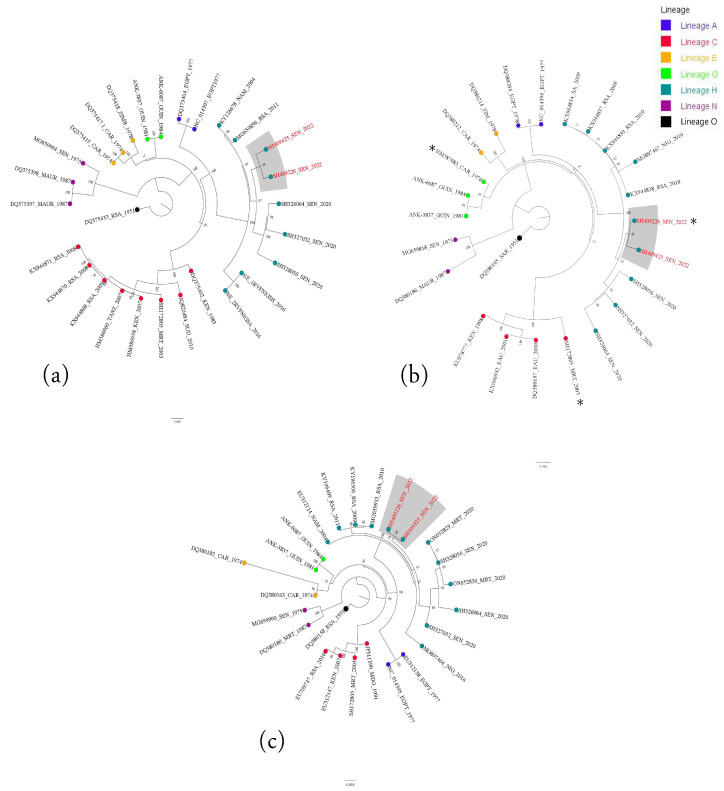
Maximum likelihood trees of RVFV strain-based L, M and S segments (panels (**a**–**c**)). The newly characterized RVF isolates are color-coded in red and highlighted in gray. The strain used in phenotypic characterization is indicated by a star (*) in (**b**) panel.

**Figure 3 viruses-16-01018-f003:**
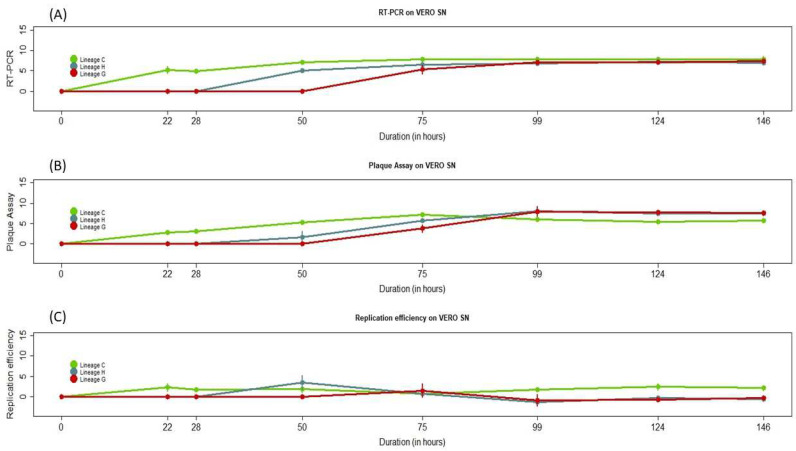
Growth kinetics of RVFV strains. Panels (**A**–**C**) show the viral RNA equivalents isolated from the supernatant (**A**) (log10 of RNA copy number), the number of infectious viral particles (**B**) (log10 PFU/mL) and the replication efficiency of RVF strains (**C**) on Vero cells over a 146 h post-infection time period. The error bars indicate the range in values of two independent experiments.

**Figure 4 viruses-16-01018-f004:**
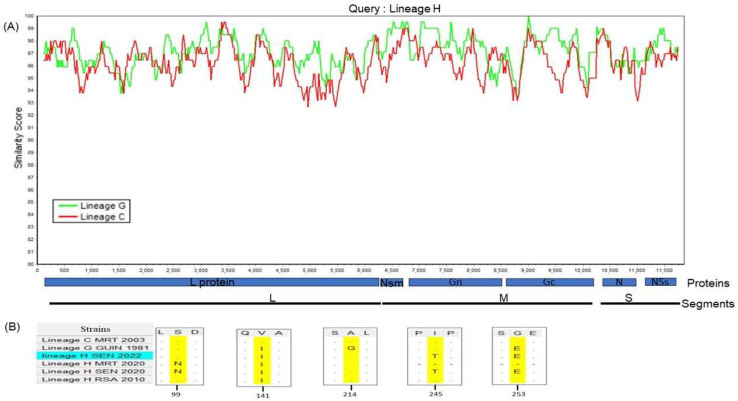
Similarity plot generated by SIMPLOT software version 3.5.1 (sliding window: 280 nt, step: 20 bp) based on the three genome segments of Rift Valley fever lineages C, G and H (**A**). The NSs protein mapping between the three lineages C, G and H. Positions are annotated and aligned to their loci in the NSs gene (**B**).

**Table 1 viruses-16-01018-t001:** Strains of Rift Valley fever virus used in this study.

Isolate	Lineage	Countries	Year	Host	Background
SH409226	H	Senegal	2022	Human	Surveillance
SH172805	C	Mauritania	2003	Human	Outbreak
ArB1976 (ZINGA)	G	Central Africa	1969	Mosquito	Outbreak

**Table 2 viruses-16-01018-t002:** Clinical profiles of the RVF cases.

Health Centers (Region)	Day of Consultation	Age (Years)	Sex	T (°C)	Signs	RVF Positive Test
**Mariste (Dakar)**	11 November 2022	32	Man	40	Headache, muscle pain, joint pain, abdominal pain, vomiting, asthenia and retro-orbital pain	RT-PCR
**Bokidiawe (Matam)**	17 November 2022	19	Woman	38.1	Headache, joint pain and muscle pain	RT-PCR
**Thilogne (Matam)**	16 August 2022	56	Man	38	Headache and muscle pain	IgM
**Podor (Saint-Louis)**	8 September 2022	35	Woman	37.8	Headache, asthenia	IgM

## Data Availability

The metadata supporting the results of this study can be obtained by contacting the authors. Due to privacy concerns for the research participants, particularly febrile patients, the data are not publicly accessible. The sequences for the L, M and S segment strains have been deposited in GenBank under the accession numbers PP541453, PP541454 and PP541455 for the SH409226 strain and PP541456, PP541457 and PP541458 for the SH409425 strain.

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
