# Peer review of "Re-Emergence of Rift Valley Fever Virus Lineage H in Senegal in 2022: In Vitro Characterization and Impact on Its Global Emergence in West Africa"

_viruses, 2024, doi:10.3390/v16071018_

Round 1

Reviewer 1 Report

Comments and Suggestions for Authors

The manuscript describes phylogenetic and phenotypic analyses of a strain of RFV isolated from 4 cases of RFV that occurred in 2022 in 3 regions of Senegal. The isolated strain belongs to lineage H, a lineage already isolated in Senegal, demonstrating the anchoring of this lineage in time and space. The phenotypic study carried out indicates that the isolated strain shows earlier and higher replication than typical strains of lineages G and C.  

This interesting study is the fruit of the sentinel syndromic surveillance network, a very important tool for identifying epidemic risks and enabling an appropriate response. However, there are a few points to be improved in the manuscript: One major point is the number of replicates (2), which is not sufficient for a statistical approach. A minimum of 3 replicates is required to reach a conclusion. A general remark also concerns the figures. In figure 2, the colors are not clearly distinguishable from one another. Furthermore, the color coding of the lineages in figure 2 should be respected in figure 3. The isolated strain should be represented by a square and not a dot like the other strains, to make it stand out more clearly.

The discussion from line 304 to line 324 needs to be reread and corrected. Some sentences are incomplete or difficult to understand. The same applies to lines 332 to 334.
One question concerns Mariste's patient. How can we explain his contamination? Did he travel to the endemic zone on the Senegalese border? This is worth discussing.
There are many errors and omissions in the rest of the manuscript, of which I shall mention a few:
Line 78, a new RVF case was reported
Line 83 All this could cause
Line 117patients are considered suspect if they present..
Line 118 headache
Line 120 filled out
Line 254: specify Zinga after lineage G
line 256 capitalize Zinga
The part of the results between line 262 and 269 should be reread and corrected: for example line 265 Lineage H and C had comparable efficiency replication linge 266 three lineages had ststistically comparable...
Line 267 to 269: unclear to be modified
Line 280 Louis

Comments on the Quality of English Language

There are several sections of the manuscript where the qulaity of the English is unsatisfactory and could be detrimental to the understanding of the text. I indicated this in my report.

Reviewer 2 Report

Comments and Suggestions for Authors

This manuscript describes an interesting study using syndromic sentinel surveillance in Senegal which included Rift Valley fever.  The study found a re-emergence of a linages H virus by phenotypic characterization.  This characterization was not well described.  Figure 2 shows a maximum likelihood analysis for L, M and S segments but no sequencing methods are described.  Growth kinetics were done in duplicate thus no statistical analysis could be performed.  The growth kinetics should be done in triplicate and statistical analysis should be done to support the authors conclusions.  This an important finding but these comments must be addressed before it is ready for publication.

Minor:

Ln 185:  what is an MOI of 10-2?  10.2 or from 2-10?  

Figure 3: not necessary could be supplementary information.

Figure 4:  What lineage each virus strain should be noted in the figure description.

Comments on the Quality of English Language

The English is fine.

Reviewer 3 Report

Comments and Suggestions for Authors

Review of “Re-emergence of Rift Valley Fever Virus lineage H in Senegal 2 in 2022: in vitro characterization and impact on its global emergence in West Africa.” (reference viruses-2967510-peer-review-v1) by Sene et al.

In this manuscript, Sene et al described four new cases of RVF detected in the northeast (Matam) and western Senegal (Dakar) using an established sentinel syndromic surveillance system network (Syndromic Sentinel Surveillance 112

System or 4S Network) as a result of surveillance efforts carried out in 2022. This repost includes molecular and phenotypic (replication kinetics) characterizations of the RVFV strains detected.

Line 35: replace “were” by “was”.

Line 37: what does exposure by ELISA mean? Please explain,

Lines 97/98: try to avoid similar words (confirmed/confirming) in two consecutive sentences.

Line 118: correct “headach” to “headache”.

Line 122: define which “biological samples” are taken from suspected RVFV-infected individuals. The se holds for “clinical sample” in line 132.

Comment: Table 1 (in the Methods section) is offset…please bring it to the middle of the page and reduce font size. The same holds true for table 2 in the Results section.

Line 162: Italicize Cercopithecus aethiops. For the sake of correction if the origin of the PS cells used is given in detail (including: ATCC number, Manassas, USA; line 163), the same rational should be used to characterize Vero cells (line 162(

Line 175: Correct “until extensive cytopathic effects.” To “until extensive cytopathic effects could be observed.”

Line 183/4/5: the 105 and 103 exponentials should be corrected (105 and 103). In line 185: correct 10-2 to 10-2. Other similar corrections are required through the document. Beware that this is NOT the notation the authors used in the legend of Fig. 3. I suggest the use of the same writing rational.

Line 187: how were the virus replication sampling times defined? The two first two are separated by 6h, before the authors decide to do sampling at approximately 24h intervals.

Lines 211/212: The sentence “will all patients recovered” is awkward. Please correct. In Line 214, correct “Figure1” to “Figure 1”. 

Comment: Given the very high identity the authors found between RVFV strains identified in 2022 and before, they suggest re-emergence of the former. Have there been any climatic or anthropogenic change that might have supported the remergence? 

Comment: Given the high genetic similarity where do the differences between the analysed RVFV strains analysed fall? Could the authors also provide a comparison of the polymorphisms (in terms of AA) between the lineage C, G and H used in the virus replication kinetics experiments?

Line 233: I suggest changing the text “In rectangle the strains used in phenotypic characterization.” To “The RVFV strains used in phenotypic characterization are indicated by the rectangles in (b).”

Comment Figure 4: Please indicate the collection points used, indicate the virus genetic Lineage by the side (or replacing) the virus strain designations. The text mentions viral lineages but they are NOT mentioned in the graph, which forces the reader to go back and forth (text/figure) to follow the authors’ reasoning.

Comment: In the methods, the authors mention having replicated RVFV for 4 days (to produce virus stocks) until extensive CPR was observed, but then choose to do their kinetics experiments using a different time-scale (up to 150h, or over 6 days)? How many times were these experiments repeated for reproducibility? Why was an moi of 10exp-2 used for these experiments? Is there any information regarding replication kinetics using different moi?

Line 269: Correct “0.005to 0.01” to “0.005 to 0.01”.

Comment: The authors suggest the possible involvement of mosquitoes in RVFV transmission (especially considering the seasonality of virus detection). Is there any possible data that might suggest which species might be implicated in this transmission? Since the biotopes and control efforts of Aedes vs Culex are different, it might help predict possible outbreaks in the future if a control of the vector population is established.

Comment: Are there Lineage-specific viral traits that set each lineage apart from the others? Is the (re)emergence of Lineage H relevant per se (associated with any specific sign, symptom, higher/lower risk of aggravated disease, etc. ? As the authors suggest Lineage H might be “less virulent” a genetic analysis of the diversity of selected viral products encoded by different viral Lineage-specific strains might be interesting to perform and add value to the paper. The authors also mention that “RVFV strains genetic diversity most likely affect viral cycle replication and affect virulence as described” (lines 330-331) but never mention whether vector competence and vector capacity changes as a function of RVFV lineage. Could this information be provided?

The FONT SIZE between 325-340 should be corrected.

Comments on the Quality of English Language

small changes have been suggested

Round 2

Reviewer 1 Report

Comments and Suggestions for Authors

The authors have taken note of the comments made by the referee and have modified the manuscript accordingly. The Results and Discussion sections are now much easier to read, and the manuscript has been improved.